

# Genetic diversity in migratory bats: Results from RADseq data for three tree bat species at an Ohio windfarm

Michael G. Sovic[1,2], Bryan C. Carstens[1] and H. Lisle Gibbs[1,2]

[1] Evolution, Ecology, and Organismal Biology, The Ohio State University, Columbus, OH, United States
[2] Ohio Biodiversity Conservation Partnership, Ohio State University, Columbus, Ohio, United States

## ABSTRACT

Genetic analyses can identify the scale at which wildlife species are impacted by human activities, and provide demographic information useful for management. Here, we use thousands of nuclear DNA genetic loci to assess whether genetic structure occurs within *Lasiurus cinereus* (Hoary Bat), *L. borealis* (Red Bat), and *Lasionycteris noctivagans* (Silver-Haired Bat) bats found at a wind turbine site in Ohio, and to also estimate demographic parameters in each of these three groups. Our specific goals are to: 1) demonstrate the feasibility of isolating RADseq loci from these tree bat species, 2) test for genetic structure within each species, including any structure that may be associated with time (migration period), and 3) use coalescent-based modeling approaches to estimate genetically-effective population sizes and patterns of population size changes over evolutionary timescales. Thousands of loci were successfully genotyped for each species, demonstrating the value of RADseq for generating polymorphic loci for population genetic analyses in these bats. There was no evidence for genetic differentiation between groups of samples collected at different times throughout spring and fall migration, suggesting that individuals from each species found at the wind facility are from single panmictic populations. Estimates of present-day effective population sizes varied across species, but were consistently large, on the order of $10^5$–$10^6$. All populations show evidence of expansions that date to the Pleistocene. These results, along with recent work also suggesting limited genetic structure in bats across North America, argue that additional biomarker systems such as stable-isotopes or trace elements should be investigated as alternative and/or complementary approaches to genetics for sourcing individuals collected at single wind farm sites.

Corresponding author
Michael G. Sovic, sovic.1@osu.edu

## INTRODUCTION

Species that migrate over large spatial scales can spend substantial amounts of time in transit between breeding and wintering locations, and experience substantial impacts from point-source anthropogenic factors during this period (*Rappole & McDonald, 1994*; *Faaborg et al., 2010*). For example, migrating bats and birds experience mortality as a

result of collisions with wind turbines and cell phone towers, habitat loss, exposure to disease, and global climate change (*Jonzén, Hedenström & Lundberg, 2007*; *Altizer, Bartel & Han, 2011*; *Loss, Will & Marra, 2013*). Identifying the scale and magnitude of these geographically limited factors can be important in assessing the scale of these impacts on species of conservation concern.

Specifically, wind turbines have been shown to cause a substantial number of mortalities to bat species (*Johnson, 2005*; *Kunz et al., 2007*), and particularly impact migratory tree bats (*Arnett et al., 2007*; *Cryan & Brown, 2007*). Thermal imaging studies demonstrate that bats may be attracted to turbines (*Horn, Arnett & Kunz, 2008*), and the low-pressure pockets of air produced by these turbines may lead to barotrama (*Kunz et al., 2007*; *Baerwald et al., 2008*). Regardless of the cause of death, a study produced by the National Renewable Energy Lab suggests that turbines will cause tens of thousands of fatalities among migratory tree bat species by the year 2020 (*Kunz et al., 2007*). Given this magnitude of potential impact, researchers and agencies are actively seeking to understand and mitigate, to the extent possible, the impact of wind farms on bat species.

Several hypotheses have been offered to explain the attraction of bats to wind energy installations (reviewed by *Kunz et al., 2007*). It may be that their insect prey are more abundant in the clearings created by these facilities, or that sound produced by the turbines could either attract or disorient the bats. Using bat detectors as monitors, recent research demonstrated that the activity of tree bats increases dramatically at tall structures during migrations for reasons other than foraging (*Jameson & Willis, 2014*). This supports earlier suggestions that bats mistake the installations for tall trees, which are likely used as lekking points by some species (particularly *Lasiurus*; *Cryan, 2008*). It may be that these facilities function as aggregation points for migratory individuals, as previous work has suggested that migratory bats are impacted more than resident populations (*Johnson et al., 2003*).

In some migratory birds, genetically distinct populations appear to use geographically restricted flyways (*Boulet, Gibbs & Hobson, 2006*; *Ruegg et al., 2014*). Thus, the impact of mortality at a given stopover site will affect a geographically restricted population, or set of populations. Whether this pattern also occurs in other migratory animals such as bats is unknown. A first step in establishing whether genetically-based biomarkers can detect similar patterns in bats is to conduct genetic assays of migratory individuals at specific sites where bats suffer high mortality. Identifying patterns of genetic structure would be consistent with multiple geographically and genetically distinct populations migrating through a single site. An association between this structure and time may further reflect temporal differences in migration patterns among any genetically distinct groups.

In addition to evaluating patterns of structure, genetic data can also be used to assess demographic trends and estimate population parameters that may be important in a conservation context. Genomic-scale SNP data, such as that generated with restriction site associated DNA sequencing (RADseq, *Davey & Blaxter, 2011*), along with novel methods that efficiently analyze large SNP datasets (i.e. *Gutenkunst et al., 2009*; *Excoffier et al., 2013*), provide powerful options for testing among alternative demographic scenarios and estimating associated population parameters. Doing so can identify patterns of

population size change over time that may reflect anthropogenic impacts, or quantify parameters such as effective population size (Ne) that may indicate a population's potential to adapt to future environmental changes, or to suffer from factors such as inbreeding depression that are related to levels genetic diversity.

*Lasiurus cinereus* (Hoary Bat), *L. borealis* (Red Bat), and *Lasionycteris noctivagans* (Silver-haired Bat) are three species of tree bats that are distributed throughout North America, and are active in the Midwest between April and November (*Kurta, 1995*). The migratory habits of these three species make them especially susceptible to effects of wind facilities, and indeed high levels of mortality have been documented for each of these species at wind farms (*Kunz et al., 2007*). Previous work using microsatellite and mitochondrial DNA data assessed genetic variation in two of these species (*L. cinereus* and *L. borealis*), and found no evidence for population-level differentiation in either group (*Vonhof & Russell, 2015*; *Korstian, Hale & Williams, 2015*). Large SNP datasets such as those described above offer an alternate source of genetic markers that may provide additional power to detect more subtle patterns of differentiation, and to estimate demographic patterns and/or parameters with greater precision. Here, we use RADseq data to determine levels and patterns of genetic diversity in samples of these three bat species collected from a wind facility in Ohio, to assess the extent to which the genetic diversity in the samples is structured by the time of death, to test for evidence of trends in population size over time, and to estimate demographic parameters using coalescent-based models.

## METHODS

### Sample collection

State agency biologists provided tissue samples from a subset of bats that were collected by the agency at an Ohio wind facility during the spring, summer and fall of 2012. Samples used for the genetic analyses presented here included 44 Hoary bats (*Lasiurus cinereus*), 37 Silver-haired bats (*Lasionycteris noctivagans*), and 53 Eastern Red bats (*Lasiurus borealis*). While samples from both *Lasiurus* species were collected throughout the season, *Lasionycteris noctivagans* were sampled either early or late in the season. A small wing biopsy was obtained from each sample and placed in 95% EtOH until further processing. DNA extraction was conducted using a Qiagen DNA Blood and Tissue kit (Qiagen, Hilden, Germany) following standard protocols.

#### Library preparation

Restriction enzymes EcoRI and SbfI were used to digest genomic DNA, and double-digest RADseq libraries were generated for each sample (*Peterson et al., 2012*). Libraries were pooled in equimolar amounts prior to sequencing, and run in single read 75-bp runs on an Illumina HiSeq 2500. Details of the library preparation methods are provided as Supplemental material.

### Bioinformatics processing

Denovo locus identification and genotyping was performed separately for each species with AftrRAD version 4.1 (*Sovic, Fries & Gibbs, 2015*), which can assemble RADseq loci in

a computationally efficient manner, while accounting for indel variation among alleles. AftrRAD analyses were carried out with default parameters, with the exception that we trimmed SNPs occurring beyond read position 55 (after also trimming the in-line barcodes and restriction recognition sequence at the beginning of reads) to remove artifactual SNPs that accumulate toward the ends of reads (see *Sovic, Fries & Gibbs, 2015*). The default parameters used include a relatively conservative threshold for initial quality filtering that removes all reads containing quality scores <20. To reduce possible effects of allele drop out (*Arnold et al., 2013*; *Gautier et al., 2013*), we conservatively retained for analyses only those loci scored in all of the samples in each dataset.

## Data analysis

### Genetic isolation by mortality time

We conducted two analyses on all three species to assess the extent to which genetic diversity was clustered as a function of the time of collection. First, we conducted an analysis of molecular variance (*Weir & Cockerham, 1984*; *Excoffier, Smouse & Quattro, 1992*; *Weir, 1996*) using Arlequin 3.5 (*Excoffier & Lischer, 2010*) in order to assess whether genetic variance was partitioned across time. Samples were divided into two (*L. noctivagans*, *L. cinereus*) or three (*L. borealis*) groups containing samples that were collected at similar times throughout the year (Fig. 1), and the genetic variance was decomposed into either among-group or within-group variance. We then compared these values to the total variance and assessed the statistical significance of our findings using randomized permutations of the data. AMOVAs are commonly conducted in order to assess the importance of geography for structuring genetic variation. Our analysis here is conducted in a similar manner, but using time rather than space as the physical axis of interest.

Second, we tested the relationship between genetic diversity and the time of sample collection in the absence of any clustering. This analysis, which we refer to as an isolation-by-time analysis, is essentially an isolation by distance analysis, but with a temporal matrix that consisted of a count of the number of days between the time of collection for any given pair of samples. A Mantel test, implemented in GenePop (*Raymond & Rousset, 1995*; *Rousset, 2008*) was used to conduct the isolation by time analysis.

In addition, we analyzed the data with STRUCTURE (*Pritchard, Stephens & Donnelly, 2000*), which performs clustering based on the genotype data alone, and can therefore identify genetic structure that may be partitioned by any factor, including time, geography, etc. For each of the three species we evaluated possible K values ranging from 1–5 (5 runs for each K value) under a model of admixture and correlated allele frequencies. The Markov chain was run for $5.0 \times 10^5$ steps per analysis, with a burn-in of $1.0 \times 10^5$. Log likelihood values were averaged across runs for each K value, and these values were evaluated along with patterns of individual assignment to clusters to draw inference regarding the optimal K value for each dataset.

## Estimation of population and demographic parameters

Finally, given that the analyses outlined above did not provide any strong evidence for genetic structure within each species (see below), we generated single population folded

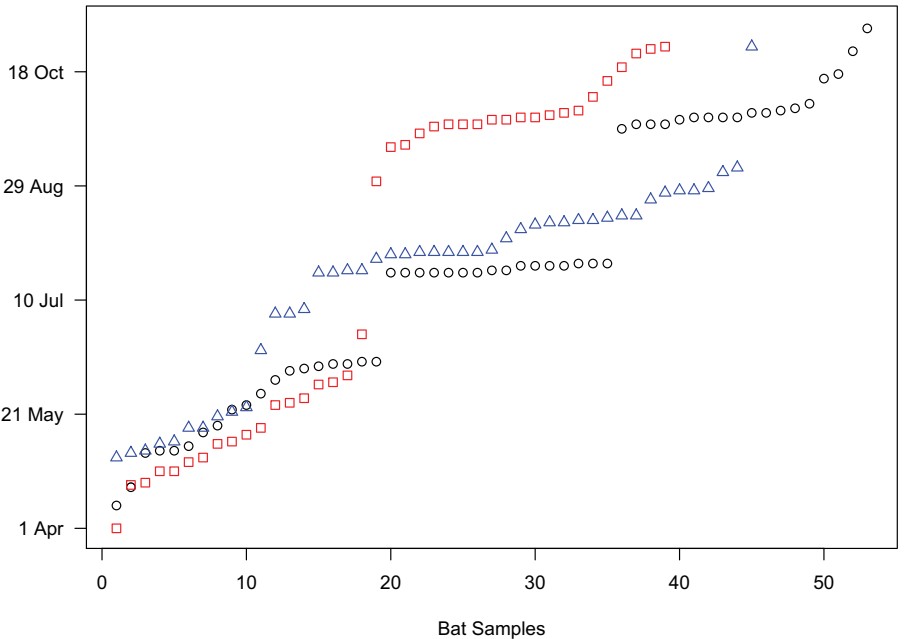

**Collection Dates for Individual Bats During 2012**

**Figure 1 Collection dates of bats chosen for genetic analyses.** Samples included in the study were selected in a manner that formed two temporal groups for *L. noctivagans* and *L. cinereus* (red squares and blue triangles, respectively) or three temporal groups for *L. borealis* (black circles). These clusters formed the basis for the AMOVA analysis.

site frequency spectra for each of the datasets using scripts associated with AftrRAD, and used these to assess historical demographic patterns and estimate associated parameters in the three species. Each single-population site frequency spectrum was generated using a maximum of one SNP per RADseq locus by using the "unlinked" flag in AftrRAD to select the first biallelic SNP at loci that contained multiple SNPs. This was done in an effort to alleviate any effects of tight linkage among SNPs at such loci. The number of monomorphic sites in the site frequency spectrum was scaled based on the proportion of linked SNPs that were removed from the dataset ("MonoScaled" flag in AftrRAD). As a sensitivity analysis, additional site frequency spectra were generated that included all sites in the dataset. For all three species, the same demographic model was chosen as optimal regardless of whether linked SNPs were included or not. Parameter estimates were also comparable between the datasets, suggesting that the conclusions related to the demographic modeling were not highly sensitive to patterns of linkage.

We used FastSimCoal252 (*Excoffier et al., 2013*) to calculate the likelihood of the observed site frequency spectra under alternative demographic models, and we compared these models with AIC. The first model represented a constant population size, and included effective population size as the only estimated parameter (Fig. S1A). The second model allowed for population growth or decline at a constant rate, followed (backwards in time) by constant population size. This model estimated the current population size, the population size prior to the growth or decline, and the time at which the growth/decline
event began (Fig. S1B). All parameters were selected from a uniform distribution of 10–1,000,000 that was not bounded on the upper end of the distribution. For each model, we performed 100 independent runs of fsc252 (30 ECM cycles, 50,000 simulations per run), and we chose the run with the highest likelihood for each model for AIC model selection. Following *Excoffier et al. (2013)*, we used point estimates for parameters from the optimal model/run for each species to perform parametric bootstrapping, and we used these bootstrapped datasets to generate confidence intervals for the estimated parameters. Parameters were estimated assuming an average rate of $2.5 \times 10^{-8}$ mutations per nucleotide site per generation, as reported by *Nachman & Crowell (2000)* for humans. To determine the sensitivity of our analyses to the value of the mutation rate parameter, we also performed model choice and parameter estimation analyses using a rate of $2.366 \times 10^{-9}$ mutations per nucleotide site per generation, which was estimated by *Ray et al. (2008)* from limited intron data for vespertilionid and miniopterid bats. The specific value of the mutation rate did not change inferences regarding model selection, but did influence parameter estimates at a magnitude comparable to the difference in the two assumed mutation rates. Specifically, effective population size estimates were approximately $10\times$ larger, and dates of historical events were approximately $10\times$ older with the lower mutation rate. We report parameter estimates based on the value of $2.5 \times 10^{-8}$ mutations per nucleotide site per generation for two reasons. First, this value leads to more conservative estimates in parameters such as Ne from a conservation perspective. Second, this value is comparable to other estimates of genome-wide mutation rates that have been calculated directly from whole-genome data (*Keightley, 2012*), suggesting that it may be a better estimate of the true mutation rate for a mammal. Effective population size estimates are reported throughout the text as the number of chromosomes in the population. As such, the number of diploid individuals represented by these numbers would be equal to half these values. Finally, overall levels of heterozygosity were calculated for each dataset using AftrRAD.

## RESULTS

### RADseq Data

Nearly 60 million sequence reads produced between ~10,800 and ~37,000 total loci per species (75 base pairs per locus, Table 1). The number of polymorphic loci scored in all samples for each of the three species was 456, 1905, and 2508, for *L. noctivagans*, *L. borealis*, and *L. cinereus*, respectively, and the total number of SNPs at these polymorphic loci was 835 for *L. noctivagans*, 12,117 for *L. borealis*, and 8,236 for. All results reported below are based on only these loci/SNPs sequenced in all individuals, as missing data, if analyzed, may lead to underestimation of genetic diversity (*Arnold et al., 2013*).

### Genetic isolation by mortality time

The AMOVA indicates that the vast majority (>99%) of the molecular variance is distributed within groups (Table 2). $F_{ST}$ values are generally low, indicating that there is little genetic differentiation among groups of samples collected at different times of the

**Table 1 Sequencing statistics.** Shown from left to right for each species are the number of samples, the average number of sequence reads per sample, the mean read depth for each locus, the total number of loci recovered, the number of total loci sequenced in all samples, and the number of polymorphic loci sequenced in all samples.

| Species | Samples | Mean reads per sample | Mean read depth | Loci (total) | Loci (scored in all samples) | Polymorphic loci (scored in all samples) |
|---|---|---|---|---|---|---|
| *L. noctivagans* | 37 | $2.49 \times 10^5$ | 11.71 | 10794 | 976 | 456 |
| *L. borealis* | 53 | $3.86 \times 10^7$ | 36.58 | 37076 | 2287 | 1905 |
| *L. cinereus* | 44 | $2.73 \times 10^7$ | 32.64 | 23920 | 3113 | 2508 |

**Table 2 Analysis of molecular variance.** Shown for each species are the source of variation, the degrees of freedom, sum of squares, variance components, and percent variation. Also shown are the $F_{ST}$ values and probability that an $F_{ST}$ of this magnitude could occur by chance. Samples are grouped into temporal clusters (see Fig. 1) that formed the basis of the among and within group comparisons.

| Species | *L. noctivagans* | | | |
|---|---|---|---|---|
| Source of variation | d.f. | Sum of squares | Variance components | Percent of variation |
| Among groups | 1 | 45.808 | 0.01371 | 0.03 |
| Within groups | 72 | 3261.665 | 45.30091 | 99.97 |
| Total | 73 | 3307.473 | 45.31461 | |
| $F_{ST}$ | 0.0003 | P(rand. value > obs. value) | | 0.64907 |
| **Species** | ***L. borealis*** | | | |
| Source of variation | d.f. | Sum of squares | Variance components | Percent of variation |
| Among groups | 2 | 828.907 | 1.48557 | 0.41 |
| Within groups | 103 | 37295.706 | 362.09423 | 99.59 |
| Total | 105 | 38124.613 | 363.57980 | |
| $F_{ST}$ | 0.00409 | P(rand. value > obs. value) | | 0.15347 |
| **Species** | ***L. cinereus*** | | | |
| Source of variation | d.f. | Sum of squares | Variance components | Percent of variation |
| Among groups | 3 | 993.182 | 1.06590 | 0.34 |
| Within groups | 86 | 26548.229 | 308.70034 | 99.66 |
| Total | 89 | 27541.411 | 309.76624 | |
| $F_{ST}$ | 0.00344 | P(rand. value > obs. value) | | 0.70674 |

year. Taken together, the results of the AMOVA suggest that there is little differentiation as a function of time in any of the three species. However, this analysis was based on an *a priori* grouping of samples, so it could be the case that methods that do not require subjective a priori clustering of samples would identify alternative partitions of genetic structure.

The probability of the data given various clustering levels (i.e., $K = 1$ to $K = 5$) was estimated using STRUCTURE, with no prior information provided in terms of the time of sampling. The overall log probability of the data was highest at $K = 1$ for both *L. borealis* and *L. cinereus* (Fig. 2), suggesting this is likely the optimal clustering solution for these

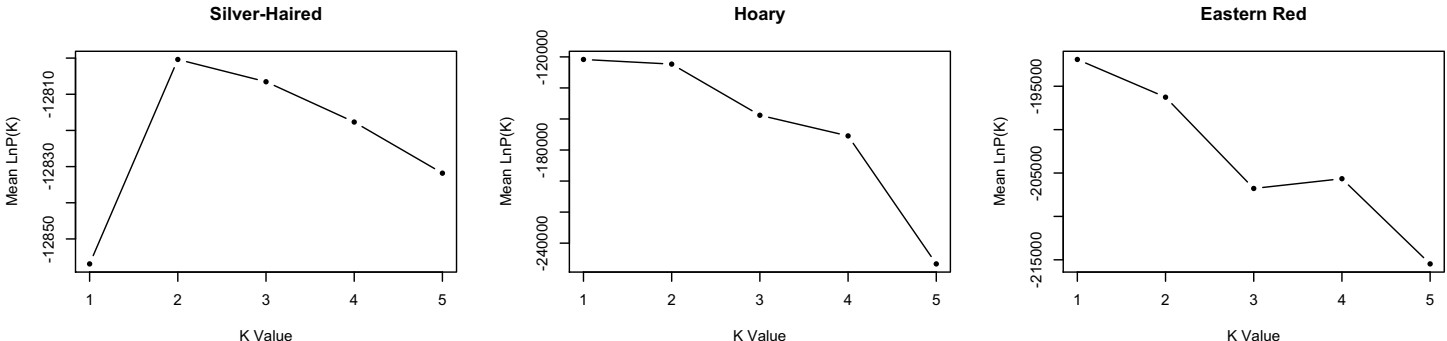

**Figure 2 Genetic clustering results from STRUCTURE.** The mean $-lnL$ for each species is shown for $k = 1$ to $k = 5$, where $k$ is the number of clusters assumed. The likelihoods for *L. borealis* and *L. cinereus* are highest at $K = 1$, suggesting these datasets may best represent single panmictic groups. The likelihood for *L. noctivagans* was highest at $K = 2$. However, individual assignment probabilities at $K = 2$ failed to show a pattern consistent with genetic structure in this species, suggesting that it too may be best characterized as a single panmictic group.

**Table 3 Isolation-by-time analysis.** Shown for each species are the slope and confidence interval of the best-fit linear relationship between genetic distance and collection time, along with the associated *p*-value.

| Species | Slope (95% CI) | *P*-value |
|---|---|---|
| *L. noctivagans* | $-7.54 \times 10^{-6}$ ($-5.73 \times 10^{-5}$, $4.92 \times 10^{-5}$) | 0.015 |
| *L. borealis* | $5.68 \times 10^{-5}$ ($-1.65 \times 10^{-5}$, $1.33 \times 10^{-4}$) | 0.002 |
| *L. cinereus* | $-1.09 \times 10^{-4}$ ($-3.05 \times 10^{-4}$, $9.65 \times 10^{-5}$) | 0.261 |

datasets. In contrast, the highest likelihood for *L. noctivagans* was at $K = 2$. However, individual assignment probabilities at $K = 2$ in *L. noctivagans* show a relatively constant amount of admixture across all samples (Fig. S2) that is not consistent with a pattern expected from structured populations. As a result, we see no strong evidence from the Structure data to reject a null hypothesis of a single genetic population in any of these three datasets.

Results from the isolation by time analysis provided little support for a relationship between genetic distance and time. While the null hypothesis of no relationship was rejected for two of the three species (*L. noctivagans* and *L. borealis*), estimated slopes were small, and 95% confidence intervals for these slopes included zero in all cases (Table 3). In addition, for the two species in which the null hypothesis was rejected based on the calculated p-value, the direction of the relationship associated with the point estimate of the slope varied, with a slightly negative relationship between genetic distance and time for *L. noctivigans*, and a slightly positive relationship for *L. borealis*.

## Estimation of population and demographic parameters
Individual runs of fsc252 under each model consistently estimated a similar likelihood, suggesting adequate searching of sample space and convergence on the maximum likelihood parameters for each model. These analyses provided strong support for a growing population in all three species, with essentially all of the relative AIC model

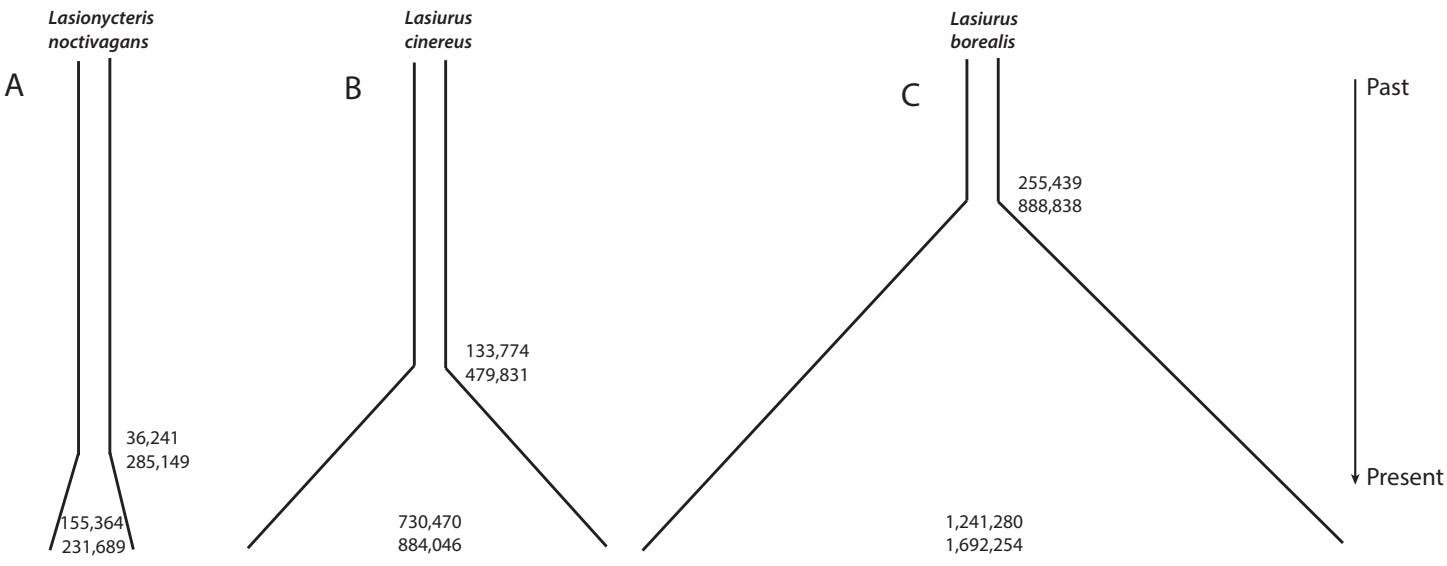

**Figure 3 Demographic parameter estimates from FastSimCoal.** Results are based on a model of increasing population size, which was identified as the optimal model for each of the three species. Numbers inside the plots represent confidence intervals calculated for current effective population sizes (number of chromosomes). Numbers outside the plots represent confidence intervals for the time at which population growth began (number of generations in the past). Figures are scaled based on the point estimates of current effective size and time since growth began to represent the relative magnitudes of these parameters across the three species.

weight assigned to a model of increasing population size in each case (Table 4, Fig. 3). *L. noctivagans* had the lowest effective population size estimates of the three species, but like the other two, was modeled as a growing population, with a current (haploid) effective population size of 189,288 (CI 155,364–231,689) that grew from an ancestral population of 7,001 (CI 11–37,656) starting 84,460 (CI 36,241–285,149) generations in the past (Fig. 3A, Table 5). For *L. cinereus*, parameter estimates suggested that population growth began 156,608 (CI 133,774–479,831) generations ago, with an ancestral effective population size of 17,023 (CI 11–40,216) growing to a current size of 830,623 (CI 730,470–884,046) individuals (Fig. 3B, Table 5). Like *L. cinereus*, estimates for *L. borealis* population sizes were relatively large, with an ancestral population of 32,957 (CI 15–84,105) individuals growing to 1,600,183 (CI 1,241,280–1,692,254) beginning 296,338 (CI 255,439–888,838) generations ago (Fig. 3C, Table 5). For all three species, estimates for the time at which population growth began correspond approximately to the early to mid Pleistocene, depending on generation length. Consistent with the trends in estimated current effective sizes from FastSimCoal252, the proportion of heterozygous loci was 0.088, 0.182, and 0.275 for *L. noctivagans*, *L. cinereus*, and *L. borealis*, respectively.

## DISCUSSION

While DNA has been successfully isolated from bat wing tissue in samples collected from wind facilities (e.g., *Korstian et al., 2013*), our study represents the first attempt to conduct next-generation sequencing on these types of samples. Our results demonstrate the feasibility of this approach, and suggest that genetic analyses of species impacted by wind

**Table 4 AIC model selection results for FastSimCoal analyses.** Two historical demographic models were compared for each of the three species: one simple model of constant population size, and a second model that allowed for a constant rate of growth or decline, the beginning of which is estimated. In all three species, the growth/decline model was strongly supported based on relative AIC model weights.

| Species/model | # Parameters | Ln likelihood | AIC | Akaike weight |
|---|---|---|---|---|
| *L. noctivagans* | | | | |
| Constant | 1 | −1556.879 | 7171.69 | $3.77e^{-31}$ |
| Growth/Decline | 3 | −1525.59 | 7031.59 | 1 |
| *L. borealis* | | | | |
| Constant | 1 | −4906.422 | 22596.01 | $4.00e^{-235}$ |
| Growth/Decline | 3 | −4671.156 | 21517.47 | 1 |
| *L. cinereus* | | | | |
| Constant | 1 | −7165.948 | 33002.41 | $3.41e^{-305}$ |
| Growth/Decline | 3 | −6860.612 | 31600.29 | 1 |

**Table 5 Parameter estimates from FastSimCoal analyses.** Parameter estimates from FastSimCoal analyses. Point estimates (CI) of demographic parameters under the optimal growth/decline model for each of the species. Effective population sizes (number of chromosomes) for the current population and for the population prior to the beginning of population growth are given by 'N_CURR' and "N_ANCES", respectively. 'T_GRO_DEC' represents the time at which population growth or decline began (number of generations in the past; a model of population growth was supported in all three species). Confidence intervals were generated by parametric bootstrapping.

| Species | N_CURR | N_ANCES | T_GRO_DEC |
|---|---|---|---|
| *L. noctivagans* | 189,288 (155,364–231,689) | 7,001 (11–37,656) | 84,460 (36,241–285,149) |
| *L. borealis* | 1,600,183 (1,241,280–1,692,254) | 32,957 (15–84,105) | 296,338 (255,439–888,838) |
| *L. cinereus* | 830,623 (730,470–884,046) | 17,023 (11–40,216) | 156,608 (133,774–479,831) |

facilities need not be limited to microsatellite or mitochondrial barcoding sequence markers (*Korstian, Hale & Williams, 2015*; *Vonhof & Russell, 2015*). We were able to genotype thousands of loci in each species across all samples, and these data were used to assess the presence of population structure, and to estimate demographic parameters in all three species.

We conducted several analyses that aimed to investigate the level of genetic structure in our data. Our results provide little support for genetic structure in the data, including genetic structure that may be partitioned as a function of the timing of mortality in our samples. For example, AMOVAs indicate that very little of the genetic variance can be attributed to samples collected at differing time points, and, similarly, clustering analyses in Structure provide little evidence for population structure in the samples. We also conducted an isolation-by-time analysis that is similar to analyses of genetic isolation by distance. In this analysis, the number of days replaces quantitative measures of distance or environmental variables. When we designed this analysis, we posited that (i) if populations were genetically structured in space, and (ii) if bats from different populations were impacted by the wind facility at different times of the year, then we might detect a signal of isolation by time. The overall lack of evidence for strong correlations between genetic distance and sampling time in each species is consistent with

our result of a lack of population structure from both the AMOVA and Structure analyses, and also with recent results reported for *L. borealis* and *L. cinereus* (*Korstian, Hale & Williams, 2015*), and separately for *L. borealis* (*Vonhof & Russell, 2015*).

*Excoffier et al. (2013)* introduced a composite likelihood method for estimating population demographic parameters using the site frequency spectrum generated from genomic datasets, and simulation testing indicates that their method (*FastSimCoal2*) provides accurate estimates of parameters given data of the type collected here. Based on the results discussed above, we modeled each dataset as a single random mating population in order to test for evidence of population size change and to estimate demographic parameters for these bats. Results for each species suggest that population size has been increasing since roughly the early-to-mid-Pleistocene, and that current population sizes are generally large.

As our manuscript was being prepared, two papers were published that also addressed the questions of effective population size and demographic trends in two of the three species investigated here. A linkage disequilibrium approach utilized by *Korstian, Hale & Williams (2015)* produced current Ne estimates for *L. borealis* and *L. cinereus* that were generally uninformative, as most confidence intervals included infinity. As they point out, linkage disequilibrium methods are not expected to be effective when true Ne values are high, as appears to be the case for these species. In contrast, our current study, and previous work by *Vonhof & Russell (2015)*, utilized coalescent approaches to generate estimates of current Ne in *L. borealis*. Despite differences in the specific methods, geographic breadth of sampling, and types of genetic markers, these two studies produced comparable estimates that were on the order of $10^5$–$10^6$ for this species. Our study is the first to provide strong evidence for large effective sizes in *L. cinereus* and *L. noctivagans*. Estimates for this latter species were the lowest of all three groups, but still exceeded $10^5$.

Estimates of demographic parameters such as Ne with model-based approaches like *FastSimCoal* are dependent on the parameters included in the model used for the estimation (*Koopman & Carstens, 2010*). By estimating parameters under different models, calculating AIC scores for each, and ranking the models using model probabilities, we have removed some of the uncertainty related to model choice from our analysis. However, additional uncertainty associated with factors such as the assumed mutation rate necessitates that the absolute values of any estimates are interpreted cautiously (see Methods for information on a sensitivity analysis related to variation in the assumed mutation rate parameter).

For all three species, a model of population growth was a better fit to the data than models of constant population size or population decline (see Table 4). *Korstian, Hale & Williams (2015)* & *Vonhof & Russell (2015)* also provided evidence for population growth in *L. borealis*, though estimates for the time at which growth began were later in these two studies (late Pleistocene) than our dates, which tend to center on growth beginning in the early to mid Pleistocene. Growth commencing at the late Pleistocene has been observed in a number of taxa, and is likely associated with geographic range expansion from Pleistocene refugia. However, our earlier dates may suggest that these bat populations are

better characterized as having undergone range shifts as opposed to range contractions during the Pleistocene. If true, this type of dynamic may have been facilitated by the high vagility of these organisms.

It is important to note that the model comparisons and parameter estimates presented here may not reflect very recent changes in population size associated with anthropogenic effects. Our ability to detect such changes may be limited, in part due to the apparently large number of individuals and effective sizes characterizing these populations. Therefore, even though no genetic data presented to date have demonstrated negative effects of anthropogenic impacts on diversity in these bats, it is possible that this is a function of detectability as opposed to reflecting a true lack of impact. As such, the data we have produced will provide a valuable baseline for possible future genetic monitoring of these populations (*Schwartz, Luikart & Waples, 2006*). Such monitoring could provide the opportunity to evaluate temporal trends in parameters that may respond at different rates to population size changes, but that require sampling across multiple time periods for inferring such trends (i.e. allelic diversity, see *Allendorf et al., 2008*).

While our study boasts a dataset that includes numbers of loci that far exceed any studies performed on these bat species to date, the geographic scope of sampling in the current study was restricted to a single Ohio wind farm. It is possible that our ability to detect population structure in our samples is simply a result of sampling from a localized area, and that more widespread geographic sampling would reveal distinct genetic populations. Work is ongoing to obtain and analyze samples from a larger geographic region. However, given the differences in the scope of sampling locations between our study and that of *Vonhof & Russell (2015)*; samples ranged from Texas north to Ontario, but didn't include Ohio, the similarities in estimates of effective population size for *L. borealis* between these two studies suggest that both may have sampled from the same genetic population. This would suggest that samples from the *Vonhof & Russell (2015)* study are unlikely to represent separate components of a meta-population structure that may have gone undetected in their analysis. In such a case, the samples from our more localized region would likely have been characterized by a smaller Ne value than those in their study that represented a larger geographic range. These results, in combination with the results addressing population structure above, generally seem to reflect a pattern of high gene flow across large geographic areas in each of these highly vagile species.

## Limits of genetics for sourcing migratory animals

In summary, we have demonstrated that RADseq can be a useful approach for generating data from large numbers of genetic loci in tree bat populations. These data allowed for inferences regarding long-term historical demographic processes, along with estimates of current effective population sizes. However, while genetic data can at times be a powerful tool for identifying the source of migratory animals (*Ruegg et al., 2014*), this requires that evolutionary processes have acted to generate differentiation between the populations of interest. Similar to other recent studies (*Korstian, Hale & Williams, 2015*;

*Vonhof & Russell, 2015*), our data show no evidence that this differentiation exists, which may be attributed to the high vagility of these species, along with large effective population sizes. Therefore, while genetic data can provide useful information on certain demographic aspects of these bats, and may be useful for long-term temporal monitoring of genetic parameters of conservation relevance, alternative methods should be investigated if management goals include sourcing samples. These alternatives may include biomarkers such as stable isotopes (*Voigt et al., 2012*) or trace elements (*Poesel et al., 2008*), and could serve as potentially powerful complements to genetic data for informing ecological and conservation questions in tree bats.

## ACKNOWLEDGEMENTS

We thank Jennifer Norris for support and for providing access to specimens, Ariadna Morales Garcia for help with collecting tissue samples, and Jose Diaz for assistance with lab work.

### Funding

This work was supported by the State Wildlife Grants Program, administered jointly by the U.S. Fish and Wildlife Service and the Ohio Division of Wildlife with funds provided by the Ohio Biodiversity Conservation Partnership between Ohio State University and the Ohio Division of Wildlife. The funders had no role in study design, data collection and analysis, decision to publish, or preparation of the manuscript.

### Competing Interests

The authors declare that they have no competing interests.

### Author Contributions

- Michael G. Sovic performed the experiments, analyzed the data, contributed reagents/materials/analysis tools, wrote the paper, prepared figures and/or tables, reviewed drafts of the paper.
- Bryan C. Carstens conceived and designed the experiments, performed the experiments, analyzed the data, contributed reagents/materials/analysis tools, wrote the paper, prepared figures and/or tables, reviewed drafts of the paper.
- H. Lisle Gibbs conceived and designed the experiments, contributed reagents/materials/analysis tools, wrote the paper, reviewed drafts of the paper.

### DNA Deposition

The following information was supplied regarding the deposition of DNA sequences:
Dryad: doi:10.5061/dryad.jh7h4.

### Supplemental Information

Supplemental information for this article can be found online at http://dx.doi.org/10.7717/peerj.1647#supplemental-information.

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
