# Peer review of "Genetic diversity in migratory bats: Results from RADseq data for three tree bat species at an Ohio windfarm"

_PeerJ, doi:10.7717/peerj.1647_

## Round 0.1 · original submission · Major Revisions

· Academic Editor

Major Revisions

This is an interesting study using bats killed at a wind turbine to understand present-day population structure and demographic history. Overall, the study appears to be carefully done, though the manuscript could benefit from some re-writing to make the results more comprehensible. Also, Reviewer 1 raises important concerns about data accessibility and completeness of the methods need to be addressed. Also, reviewer 2 has concerns about spatial sampling. These need to be dealt with in a future revision.

Reviewer 1 ·

Basic reporting

Sovic et al. do a careful job of evaluating population genetic structure of three species of tree bats, and perform some interesting analyses of the relationship between population structure and time of sampling. There is also an opportunity to compare population size estimates to other studies that employ microsatellites and mtDNA, however more complete information on demographic modeling methods and model fits to data would bolster the implication that the population sizes estimated in this ms are better than others.

Introduction: The goals of testing for population structure and differentiation as a function of time are well justified, but an introduction or justification for determining effective population sizes of each species would provide better context for this analysis. (For example, is there a conservation context? Is the goal a comparison of different genetic approaches?)

Supplementary data: Genetic data provided (or soon to be provided) appear complete, however the time of sampling of each individual is not provided, so that information necessary to replicate the study is not complete.

Supplementary figures: I could not find labels for the supplementary figures.

Experimental design

Methods: More detail on how you ran fsc252 is necessary. For example, did you use all SNPs in the analysis? If so, how did you deal with linkage? Did you assume each RAD locus independent? Each SNP? Did you specify recombination/mutation rates? If so, where did you get them? What were parameter bounds? How would potential linkage between markers affect model selection? (I realize that differences between AIC scores of alternative models were large, but including a brief discussion here of how linkage affects model selection might be relevant.) Either way, more detail on exactly how the fsc252 runs were conducted is needed, especially considering this analysis is a major focus of the discussion.

Validity of the findings

Figure 3 is not particularly useful, since the model is pretty simple. (If you do want to keep it, scaling it to population size would make it slightly more informative.)

It would be very useful to include a figure of the SFS for observed vs. simulated data for each model type, which would presumably demonstrate the better fit of the data to the population growth model, and give the reader a sense of how well the data fit the model overall for each species.

Lines 276-291: Your conclusion of post-glacial expansion based on the comparison of growth versus constant population models is convincing, as is your suggestion that the use of thousands of markers results in narrow confidence intervals. However, you suggest that point estimates from previous studies underestimate Ne, but do not provide enough information on how you estimated Ne (e.g. detailed fsc252 methods, mutation rate, etc.) to adequately convince the reader that your estimate of Ne is better. Differences in point estimates are influenced by differences between demographic modeling methods, mutation rates and substitution models, and your argument would be much improved by allowing a more complete evaluation of your methods.

Additional comments

Line 34: “Large numbers of loci (1000’s)” could simply be “Thousands of loci”

Lines 191-195 and Table 1: not clear if loci are RAD loci or bases, make clear that these are loci, and maybe report number of SNPs too.

Lines 207-211: This is a bit confusing. Why use the Evanno et al method at all if it will be disregarded anyway because K=1 is plausible?

Lines 285 and 288: Providing confidence intervals here for Ne estimated in this analysis would be helpful to compare with other studies without having to refer to the table.

Reviewer 2 ·

Basic reporting

The manuscript by Sovic et al. presents RADseq data from three bat species (L. cinereus, L. borealis, and Lasionycteris noctivagans) collected from Ohio. The authors do an excellent job presenting their data and the manuscript is well written. The statistical analyses that were preformed are adequate and the authors address each of the goals listed within the abstract. Here are my suggestions:

1) The authors should upload their raw data to the NCBI SRA repository and provide the project accession number within the manuscript. Public access to these data will be useful for the scientific community. I could not identify within the main text whether or not the authors uploaded the Illumina data to the SRA.

2) I was unable to access the supplementary methods. Have the authors described the quality filtering of the Illumina data within the supplementary material? Exactly how were the Illumina fastq data filtered prior to data analysis (what was the minimum quality score threshold, what trimming parameters were used, were these paired end data)?

3) What exactly is Figure 1 showing? Why isn’t the X-axis labeled?

4) The sample size for the study is relatively large, but why didn’t the authors incorporate a few individuals from outside of the wind power facility in northwestern Ohio? If the authors would have included a few outlier samples from other distant geographic areas they could have seen whether or not the RADseq data was powerful enough to identify these as outliers (or if the populations were truly panmictic). This could have been relatively easy to do, as several natural history museums most likely have tissues from these species available upon request. Incorporating individuals from geographically distant areas would have strengthened the study and might have helped to determine origin. This needs to be stated in the “Limits of genetics for sourcing migratory animals” section. Identifying the source of migratory animals requires a robust experimental design that incorporates individuals from multiple geographic areas, ideally across the range of the species. Along this same line, what did the authors do with the bats? Were they deposited in a natural history museum? If not, why not?

5) The Discussion could be reorganized slightly to make it more powerful. The authors have an opportunity to discuss what the genetic data (collectively) informs us about the biology of the study species. The authors cite Koristian et al. 2015 and Vonhof and Russell 2015 and summarize these works, but the way this is done lacking some how. The study species are migratory, and the origin of the species is uncertain, however a map of exactly where these individuals were collected in relation to the individuals sampled by Koristian et al, Vonhof and Russell, etc. would be useful. This relates to the comments I made point #4.

Experimental design

The sample size for the study is relatively large, but why didn’t the authors incorporate a few individuals from outside of the wind power facility in northwestern Ohio? If the authors would have included a few outlier samples from other distant geographic areas they could have seen whether or not the RADseq data was powerful enough to identify these as outliers (or if the populations were truly panmictic). This could have been relatively easy to do, as several natural history museums most likely have tissues from these species available upon request. Incorporating individuals from geographically distant areas would have strengthened the study and might have helped to determine origin. This needs to be stated in the “Limits of genetics for sourcing migratory animals” section. Identifying the source of migratory animals requires a robust experimental design that incorporates individuals from multiple geographic areas, ideally across the range of the species. Along this same line, what did the authors do with the bats? Were they deposited in a natural history museum? If not, why not?

Validity of the findings

The authors do a good job of describing their results. The statistical analyses were adequate and the findings are valid. The raw data should be uploaded to the NCBI SRA archive (or similar). The authors should provide the accession number in the main body of the manuscript.

Additional comments

see comments above

---

## Round 0.2 · Minor Revisions

· Academic Editor

Minor Revisions

I feel that the authors did a good job with the manuscript revisions, and it was a strong manuscript to begin with. I would suggest using the bat mutation rate suggested by the reviewer for greater accuracy.

Reviewer 1 ·

Basic reporting

No comments

Experimental design

No comments

Validity of the findings

No comments

Additional comments

The authors did a good job in revising the manuscript. The methods are much clearer and the discussion is more relevant to the goals of the paper.

On a minor note, there is a neutral mutation rate estimate for bats in Ray et al. (2008) Genome Res. 2008 18: 717-728. If the authors choose to keep the human mutation rate estimate, please provide units for the rate (i.e., per bp per generation).

---

## Round 0.3 · accepted · Accept

· Academic Editor

Accept

The authors have successfully dealt with the minor revisions requested by the reviewer, mainly incorporating an estimate of mutation from bats. The revised manuscript has addressed all outstanding issues and is suitable for publication.